# Developmental Dynamics of the Gut Virome in Tibetan Pigs at High Altitude: A Metagenomic Perspective across Age Groups

**DOI:** 10.3390/v16040606

**Published:** 2024-04-14

**Authors:** Runbo Luo, Aohan Guan, Bin Ma, Yuan Gao, Yuna Peng, Yanling He, Qianshuai Xu, Kexin Li, Yanan Zhong, Rui Luo, Ruibing Cao, Hui Jin, Yan Lin, Peng Shang

**Affiliations:** 1College of Animal Science, Tibet Agricultural and Animal Husbandry University, Linzhi 860000, China; luorunbo@xza.edu.cn (R.L.); lkx082623@163.com (K.L.); 18225165833@163.com (Y.Z.); 2College of Veterinary Medicine, Nanjing Agricultural University, Nanjing 210095, China; crb@njau.edu.cn; 3State Key Laboratory of Agricultural Microbiology, Huazhong Agricultural University, Wuhan 430000, China; gah2020hzau@webmail.hzau.edu.cn (A.G.); mabin1996@webmail.hzau.edu.cn (B.M.); hz_gaoyuan@webmail.hzau.edu.cn (Y.G.); pyn@webmail.hzau.edu.cn (Y.P.); 2021302110171@webmail.hzau.edu.cn (Y.H.); xqs1@webmail.hzau.edu.cn (Q.X.); luorui@mail.hzau.edu.cn (R.L.); jinhui@mail.hzau.edu.cn (H.J.); 4College of Animal Medicine, Huazhong Agricultural University, Wuhan 430000, China; 5College of Animal Science and Technology, Nanjing Agricultural University, Nanjing 210095, China

**Keywords:** viral metagenomics, Tibetan pig, gut virome, vAMG, phage lifestyle

## Abstract

Tibetan pig is a geographically isolated pig breed that inhabits high-altitude areas of the Qinghai–Tibetan plateau. At present, there is limited research on viral diseases in Tibetan pigs. This study provides a novel metagenomic exploration of the gut virome in Tibetan pigs (altitude ≈ 3000 m) across three critical developmental stages, including lactation, nursery, and fattening. The composition of viral communities in the Tibetan pig intestine, with a dominant presence of *Microviridae* phages observed across all stages of development, in combination with the previous literature, suggest that it may be associated with geographical locations with high altitude. Functional annotation of viral operational taxonomic units (vOTUs) highlights that, among the constantly increasing vOTUs groups, the adaptability of viruses to environmental stressors such as salt and heat indicates an evolutionary response to high-altitude conditions. It shows that the lactation group has more abundant viral auxiliary metabolic genes (vAMGs) than the nursery and fattening groups. During the nursery and fattening stages, this leaves only *DNMT1* at a high level. which may be a contributing factor in promoting gut health. The study found that viruses preferentially adopt lytic lifestyles at all three developmental stages. These findings not only elucidate the dynamic interplay between the gut virome and host development, offering novel insights into the virome ecology of Tibetan pigs and their adaptation to high-altitude environments, but also provide a theoretical basis for further studies on pig production and epidemic prevention under extreme environmental conditions.

## 1. Introduction

The Tibetan pig, endemic to the Chinese plateau region, exhibits distinct biological traits that facilitate its acclimatization to the harsh environment of the Tibetan Plateau, characterized by low oxygen levels, cold temperatures, high altitudes, and heightened radiation. Serving as a consistent meat source for plateau inhabitants, this breed has evolved into a fundamental economic cornerstone within plateau-based agriculture [1,2]. The gastrointestinal microbiota of the Tibetan pig is pivotal for its health, profoundly influencing its growth and overall well-being. Genomic-level cluster analyses have elucidated that Tibetan pigs, adapted to this elevated terrain, exhibit marked differences in genetic architecture compared to other porcine breeds. This indicates a potential interplay between their unique genetic attributes and the composition as well as functionality of their gut microflora [3,4]. The realm of fecal virome delves into characterizing intestinal viral communities and their implications in host health and pathogenesis, shedding light on the role of viruses in animal development, health, and their possible association with diseases [5,6]. The 16S rRNA amplicon sequencing technique, an established tool for probing microbial community structures, offers an exacting approach to analyze the gut microbiota of Tibetan pigs. Recent investigations using 16S rRNA amplicon sequencing of the porcine cecum microbiota have unveiled disparities between Tibetan and PIC (lean) pigs. Specifically, the Tibetan pigs showcased an augmented presence of *Bacteroidetes* and *Spirochaetota*, associated predominantly with cellulose degradation, while the PIC pigs displayed a higher abundance of *Proteobacteria*, chiefly characterized by *Campylobacter* and *Helicobacter* [3]. Through amplicon sequencing analysis of the colonic contents of Tibetan and Yorkshire pigs, it was discerned that the bacterial community within the colon of the Tibetan pigs exhibited greater richness compared to the Yorkshire pigs. Moreover, the relative abundance of *Lactobacillus* and *Bifidobacterium*, genera known for their association with enhanced disease resistance, was markedly elevated in the Tibetan pigs [7]. Concurrently, research indicates an upregulation of gut microbial genes associated with propionic acid metabolism and unsaturated fatty acid biosynthesis in Tibetan pigs residing on elevated terrain. This underscores the potential adaptive significance of the distinct intestinal microbiota of Tibetan pigs to high-altitude conditions [8]. However, a knowledge void persists concerning the intestinal virology of Tibetan pigs, necessitating comprehensive exploration in this domain.

Weaning, a crucial phase in the growth cycle of pigs, involves a switch from lactose to plant-based carbohydrates in the diet [9]. The incorporation of dietary fiber leads to alterations in the catabolic pathways of piglets, ultimately impacting the abundance and diversity of their gut microbiota [10,11]. In a study focusing on the virome of wasting piglets aged 0 to 9 weeks, it was discovered that weaning facilitated the transition of pathogenic viruses from enterovirus G and rotaviruses to porcine sapovirus [12]. In addition, Smoľak et al. identified that weaners possess fifteen viral genera significantly higher than the four found in fattened pigs using viral metagenomics [13]. In a separate investigation, Sachsenröder and colleagues demonstrated that the proportion of porcine viruses in the intestinal virome decreased with increasing pig age, whereas the opposite was observed for phages [14]. This suggests a potential correlation between the porcine enteric virome and age [15,16]. Nonetheless, there is still a gap in the research on the intestinal virome of weaned Tibetan pigs.

Whether a mild phage infects a host via a lysogenic or lytic cycle depends on the environment; stressors such as DNA damage or nutritional deficiencies can induce prophages to enter the lytic cycle, with phages favoring the lysogenic cycle in the intestinal tracts of healthy young adults, whereas, in the intestinal tracts of centenarians, concomitant inflammatory stimuli and nutritional deficiencies may induce phages to switch more frequently to the lytic cycle [17]. Host abundance also influences the mode of infection; when bacteria are more abundant, phages prefer to infect bacteria via the lytic cycle. Host abundance also affects phage infection, with phages preferring to infect bacteria via the lysogenic cycle when bacterial numbers are high. In the late stages of infection, when the amount of bacteria decreases and the phage faces the risk of not finding a new host, it will switch to the lysogenic cycle state, coexisting with the host for a long time and retaining the ability to reproduce [18]. In extremely harsh environments such as the deep sea, phages are more likely to survive by infecting the host through the lysogenic cycle [19]. In the normal intestine, phages generally integrate their genomes into bacteria via the lysogenic cycle [20,21]. 

Phages have the ability to encode viral Auxiliary Metabolic Genes (vAMGs), which play an important role in phage–host interactions. In recent years, an increasing number of studies have revealed the important role of vAMGs in altering metabolic and biochemical processes. In 2015, Maaroufi and Levesque identified a vAMG in Bacillus subtilis phages that encodes glycoside hydrolase family 32 [22], which may be involved in the metabolism of fructose oligosaccharides (FOS), further improving gut function and repairing gut damage [23]. Mihnea R. et al. found that phages in the gut of individuals at high risk of developing rheumatoid arthritis disease can encode unique vAMGs associated with anti-cyclic citrullinated protein antibodies and carry clusters of transferase enzymes that can affect bacterial cell wall polysaccharide and biofilm formation, thereby participating in immune evasion and influencing immune regulation and disease progression [24].

This paper aims to fill the research gap in the study of the intestinal virome of Tibetan pigs at different age groups. Through conducting viral metagenomic and bioinformatic analyses on Tibetan pigs from various age groups in the Tibetan region, the study uncovers the relationship between the diversity of intestinal viruses and the growth and development of Tibetan pigs. Additionally, through comparing the distribution and diversity of porcine intestinal viruses among different samples and predicting host information for intestinal phages, this study uncovers differences in the composition of intestinal viruses and the distinct lysogenic strategies employed by phage hosts. This research represents the first systematic analysis of the characteristics of intestinal viruses in Tibetan pigs at different developmental stages, contributing to filling a critical gap in this field. It offers a theoretical foundation for the production, breeding, and feed compatibility of Tibetan pigs and presents new avenues for more in-depth research.

## 2. Materials and Methods

### 2.1. Animal Management and Sample Collection

The experiments were conducted in a randomized block design with 5 piglets at the stage of lactation (L), 5 pigs at the stage of nursery (N), and 5 pigs at the stage of fattening (F). Pigs were housed with 5 pigs per pen divided into 3 blocks. Pigs had free access to food and clean water.

In this study, five fecal samples were collected from pre-weaned piglets, five fecal samples were collected from the floor of the pens on day 7 of weaning of nursery pigs, and five fecal samples were collected from fattening pigs. To avoid contamination of the fecal samples, the floor of the pens was cleaned prior to sampling, and the sampling time was 1–2 h of peak defecation per day or morning feeding. Fecal samples were stored at −20 °C for one week. Our experiment was approved by the Institutional Animal Care and Use Committee of Huazhong Agricultural University to meet the requirements of animal ethical welfare (HZAUSW-202400021).

### 2.2. Metagenomes Collection and Sequencing

Fifteen samples were enriched by the precipitation enrichment method, and 1.5 g of fecal samples were separately suspended in 15 mL of sterile PBS buffer, then subjected to vigorous oscillatory mixing for 5 min. The mixture was subsequently incubated at a temperature of 4 °C for 30 min. The suspension was subjected to centrifugation at 4 °C and 4500 rpm for 10 min to separate the larger particles. Subsequently, the supernatant was transferred to a new EP tube and centrifuged again under the same conditions. Afterwards, filter the supernatant through a 0.45 µm PVDF membrane (Millipore, USA) to exclude eukaryotic, bacterial, and cell-sized particles. The precipitate was subjected to ultracentrifugation at 4 °C and 180,000× *g* for 3 h. Then, it was resuspended in 400 μL of sterile PBS and incubated with 8 U of TURBO DNaseI (Ambion, USA) and 20 U of RNase A (Fermentas, Canada) for 30 min at 37 °C.

Subsequently, viral nucleic acids were extracted using the QIAamp MinElute Virus Spin Kit (Qiagen, Germany) in accordance with the manufacturer’s instructions and eluted into RNase-free water. The resulting high-quality viral DNA samples (OD260/280 = 1.8 to 2.2, OD260/230 ≥ 2.0) were used to construct sequencing libraries for next-generation sequencing.

Metagenomic libraries were prepared following TruSeqTM Nano DNA sample preparation Kit from Illumina (San Diego, CA, USA), using 1 µg of total DNA. DNA end repair, A-base addition, and ligation of the Illumina-indexed adaptors were performed according to Illumina’s protocol. Libraries were then size-selected for DNA target fragments of ~400 bp on 2% Low-Range Ultra Agarose followed by PCR amplification using Phusion DNA polymerase (NEB) for 15 PCR cycles. All samples were sequenced by the Illumina NovaSeq 6000 platform (150 bp × 2, Shanghai Biozeron Biotechnology Co., Ltd., Shanghai, China).

### 2.3. Metagenomic Assembly

The quality control of metagenomic raw reads was performed using the Trimmomatic v0.36 [25] with default parameters. The sequence data containing the host genome were then removed by bowtie2 [26,27,28], resulting in high-quality viral DNA reads for subsequent analysis. 

Meanwhile, the high-quality reads were assembled utilizing MegaHit v1.1.1 [29] with the parameter “--min-contig-len 200” to generate contigs for each sample. Afterwards, the contig with a length exceeding 2000 bp was chosen to identify the virus through IMG/VR v4, VirFinder v1.1, and VirSorter2 v2.0 [30,31,32] tools with the database.

### 2.4. Virus Operational Taxonomic Unit (vOTU) Clustering and Annotation

All curated viral contigs with an identity of ≥95% and a coverage of ≥85% were dereplicated using Mummer software [33]. The longest contig sequences for each cluster were chosen as representative sequences for the corresponding vOTUs. To verify the novelty of the vOTUs, the vOTUs sequences were compared with the IMG/VR virus database through blastn v2.0.6 software [34]. Following this, the species identification was established by geNomad software v1.8.0 [35], and PhaBOX (https://phage.ee.cityu.edu.hk/) [36] was employed to predict the host information of the phage. The taxonomic level of the sample was determined using the geNomad results, including phylum, class, order, and family. Virus–host correlation was subsequently analyzed using the PhaBOX results.

After the initial analysis of vOTUs, functional annotation of vOTUs and their corresponding contigs was required. The prediction of open reading frames (ORFs) for vOTUs was undertaken through METAProdigal v2.6.3 [37]. A search was then conducted on the gene set, and the Kyoto Encyclopedia of Genes and Genomes (KEGG) database was utilized with kofam v1.2.0 [38] to recognize proteins and obtain their functional annotations. Additionally, structural and functional annotations were obtained through BLASTP-based searches employing DIAMOND (v0.9.22.123) [39] against Gene Ontology (GO) database, NCBI NR database, Swiss-Prot database [40], and eggNOG v5.0 [41].

### 2.5. Fluctuation Trend of Variables and Function Identification of vOTU

To assess the variations in viral operational taxonomic units (vOTUs) across different growth stages of piglets, we analyzed dissimilarities by assessing coverage values. The coverage value could be counted as the sequence length of vOTUs covered by reads in each sample as a proportion of the total length of vOTUs. Kruskal–Wallis tests were used to appraise the distribution of every vOTU among the three groups. In case of significant dissimilarities, further Dunn’s tests were conducted to ascertain pairwise significance. The vOTUs were classified into six categories: singleton, constant, continuous growth, initial growth followed by decline, continuous decline, and initial decline followed by increase. An enrichment analysis was conducted on the annotated vOTUs outcomes, with an emphasis on retaining q-values.

### 2.6. The Identification and Analysis of Viral Auxiliary Metabolic Genes (vAMGs)

In order to identify putative viral auxiliary metabolic genes (vAMGs) that may facilitate host adaptation to the environment, DRAM-v [42] workflow was performed on viral contigs using default parameters. As required, CheckV v0.7.0 [43] was employed to eliminate vOTUs in regions with host contamination, and then all the identified vOTUs were annotated through VirSorter2 (--prep-for-dramv) to produce the affi-contigs.tab file. DRAM-v assesses the presence of viral genes both upstream and downstream of the putative vAMGs, generating an ‘auxiliary score’ to measure the confidence of the vAMGs prediction. By default, a gene is considered a potential vAMG if the auxiliary score is less than 4, has been assigned a metabolism flag (M), and has not been assigned an attachment flag (A), viral flag (V), or transposon flag (T). Those without gene IDs or gene descriptions were also discarded, and annotation matches for KEGG were present for all retained vAMGs. Following the previous reports [44,45,46], we manually removed specific metabolic pathways that would be expected in viruses, including nucleotide metabolism, glycosyl transferases, and ribosomal proteins.

### 2.7. Methods of Lysogenic and Lytic Lifestyle Identification

Phage lifestyle prediction was performed using BACPHLIP v0.1 [47] and PHATYP v3.0 [48] software. The input sequences were ensured to be annotated as phages by geNomad. According to the different needs between the software, the BACPHLIP input sequences were also verified as complete by CheckV v0.7.0. Results labelled as low-quality or unclassified were excluded and the relative abundance difference between lysogenic and lytic was counted between each group.

### 2.8. Statistics Analysis

R packages vegan (version 2.6.2) and tidyverse (version 1.3.1) were used to calculate alpha diversity (Richness, Shannon, and Simpson) to reflect the diversity of microbial communities. The Kruskal–Wallis test was employed to test for differences between the groups. Additionally, the beta diversity was evaluated by implementing non-metric multidimensional scaling analysis (NMDS) and principal coordinate analysis (PCOA). One-way ANOVA calculations and plotting for species between groups were analyzed using prism 8.0; the rest of the plotting was completed via ggplot2.

To describe the relative abundances of viruses, TPM (Transcripts Per Kilobase of exon model per Million mapped reads) values were used. The RPK (Reads Per Kilobase) values of contigs were calculated, and each contig was standardized. Quality-controlled reads from each sample were mapped to viral operational taxonomic units (vOTUs) using BWA MEM (https://github.com/lh3/bwa, default parameters). To standardize the sequencing depth, the number of reads was divided by the total number of reads for each sample [49]. The equation is as follows:$$PM=\frac{N_{i}}{L_{i}}\times 10^{6}/\sum_{1}^{n}\left(\frac{N_{i}}{L_{i}}\right)$$
where $N_{i}$ epresents the abundance of each vOTU, $L_{i}$ epresents the length of each vOTU, and $\sum_{1}^{n}\left(\frac{N_{i}}{L_{i}}\right)$ epresents the sum of values normalized by length.

## 3. Results

### 3.1. Sequencing Data and Quality Control

In this study, fecal samples were collected for viral metagenomics sequencing from five 15-day-old unweaned piglets, five 21-day-old post-weaning nursery pigs, and five 150-day-old fattening pigs. The sequencing data volume of each sample was 10 G, and a total of 150 G of original data were generated. Through the quality control of the sequencing quality of the original data and removal of the host genome fragment, it was found that the sequencing positive rate of the viral metagenomics was greater than 85% and the library construction quality was good (Appendix A).

### 3.2. Assemble of Metagenomic Data and vOTUs Construction

Considering the potential for false-positive errors arising from read-based categorical annotations, we employed the Megahit algorithm for the assembly of quality-filtered reads in this investigation. Pertinent assembly metrics, as delineated in Appendix A, will inform subsequent annotation analyses with heightened precision. Notably, the assembled metagenomic datasets in this investigation exhibited commendable contiguity, with an N50 exceeding 1000 bp. Concurrently, the average assembly across the datasets revealed in excess of 30,000 viral samples, underscoring the robustness of the dataset for ensuing viral taxonomic analyses.

To enhance the precision of the viral sequence annotations, we obtained potential viral sequences through IMG/VR, VirFinder, and VirSorter2. A total of 46,562 contigs were retained (Appendix A). During sequence alignment via the MUMmer software v2.1 suite, the sequences were recognized as a single vOTU and were amalgamated if they displayed a sequence similarity greater than 95% and the aligned length constituted over 85% of the complete sequence length. Implementing this integration criterion yielded 40,433 vOTUs with a mean length of 4898 bp. Subsequently, the blastn software v2.9.0 was employed to align the 40,433 vOTUs with the IMG/VR virus database. Thresholds of ≥90% similarity and ≥5% coverage were set to delineate “reported viruses”; sequences failing to meet these criteria were classified as “unreported viruses”. This analysis resulted in the identification of 2318 “reported virus” fragments within the Tibet pig gut virome, corroborating the annotation outcomes derived from the three aforementioned tools.

### 3.3. Taxonomy Composition of Tibetan Pig Gut Virome

The taxonomy annotation from the geNomad software is shown in Figure 1A,B. The viruses belonging to the *Microviridae* family demonstrated an abundance exceeding 57.7% across all 15 samples, positioning them as the most predominant viral entities in this dataset. 

Alterations in Predominant Microbial Families: We employed one-way ANOVA to elucidate the dynamics within the major microbial families. As illustrated in Figure 1C, pronounced differences (*p* < 0.05) were detected in the *Petitvirales* Unclassified, *Cirlivirales* Unclassified, *Caudoviricetes* Unclassified, and *Mulpavirales* Unclassified families. Specifically, within the *Petitvirales* Unclassified family, the expression level of the lactation group markedly surpassed that of the nursery group and fattening group. Similarly, in the *Cirlivirales* Unclassified family, the expression of the lactation group was significantly elevated compared to the fattening group. Furthermore, within the *Mulpavirales* Unclassified family, the lactation group exhibited a significantly enhanced expression level relative to both the nursery and fattening groups.

Alterations in Dominant Viral Phyla: Comprehensive analysis of the major microbial phyla revealed pronounced discrepancies within *Uroviricota*, as presented in Figure 1D (*p* < 0.05).

### 3.4. Analysis of Diversity Differences between Samples Based on vOTUs

To discern the variations among the samples, we computed the Richness, Shannon, and Simpson diversity indices. We depicted the comparative analysis of these parameters across samples employing box-and-line plots; the ensuing illustrations elucidate the results comprehensively. As inferred from the box plots (refer to Figure 1E), there is a discernible trajectory wherein the viral diversity in the intestinal tract of Tibetan pigs initially escalates and subsequently attenuates as the pigs advance in age. Such a trend suggests a conceivable correlation between the intestinal viral consortium of Tibetan pigs and their developmental phase.

Additionally, beta diversity analysis was employed to discern inter-sample disparities in viral diversity. Segmented by respective age brackets, this study harnessed PCoA (principal coordinate analysis) and NMDS (non-metric multidimensional scaling analysis) for a more granular quantitative assessment. As depicted in Figure 1F, the samples from distinct age brackets manifested salient clustering tendencies within both the PCoA and NMDS analytical frameworks, categorically partitioning them into the pre-weaned piglet, nursery pig, and fattening pig clusters. Consistent with the results of the Richness, Shannon, and Simpson diversity indices, this further confirms that the diversity of viruses in the intestinal tracts of Tibetan pigs of different ages differed significantly and may be related to their growth and developmental processes.

### 3.5. Host Prediction of Tibetan Pig Gut Virome

Host prediction: To better understand the presence of viruses in the intestine, we predicted the hosts of the virus by PHABOX. As shown in Figure 2, the Sankey plot reveals the dominant hosts of the five phages. The three highest-ranking bacterial genera are *Lactobacillus*, *Streptococcus*, and *Flavobacterium*, with the species including *Lactobacillus fermentum*, *Streptococcus gordonii*, and *Flavobacterium columnare*.

### 3.6. Functional Annotation and Functional Enrichment Analysis of vOTUs

Given the pronounced disparities in the viral diversity within the gastrointestinal tracts of Tibetan pigs across three distinct age brackets, and the correlations with their growth and physiological maturation, our investigation delved deeper into this observation at the gene functional dimension. To achieve this, we employed the METAProdigal tool for the genetic annotation of all the vOTUs fragments, leading to the identification of 287,705 coding sequences. Subsequently, these inferred gene sequences underwent blastx alignment against the NR, Swiss-Prot, eggNOG, KEGG, and GO databases, with the intent of gleaning functional insights pertaining to the respective viral genes. As demonstrated in Appendix A, functional annotations based on these five different protein databases successfully provided annotations for 32.78% of the vOTU genes, providing a database for further comparative analysis.

To elucidate the potential functional ramifications of vOTU genes on Tibetan pigs, this study leveraged the COG, GO, and KEGG databases for a comprehensive functional annotation analysis of the annotated vOTU genes (Figure 3A–C). As illustrated in Figure 3A, the annotation analysis using the COG database revealed a total of 26 functions. The predominant functions encompassed aspects of viral replication, recombination, and repair-related functional proteins. These were closely trailed by functionalities pertinent to the biosynthesis of cellular membranes and envelope structures. Such findings suggest a strong functional affinity of the vOTU genes primarily with biometabolic processes.

In the subsequent phases of this study, we utilized the KEGG database to annotate the signaling pathways in which the vOTU genes are involved, and to delve into their specific functional mechanisms. As illustrated in Figure 3B, annotation via the KEGG database yielded 20 pivotal signaling pathways. Among these, the metabolic pathway emerged as the most enriched, with 2800 hits, trailed by the nucleotide metabolic pathway (1592), pyrimidine metabolic pathway (1562), DNA replication pathway (1051), and homologous recombination (992). These findings align with the results from the COG analysis, reinforcing the notion that vOTUs’ genes predominantly engage in processes related to viral replication and metabolism.

In the concluding analyses, the GO database was employed to functionally enrich the biological processes, molecular functions, and cellular localizations associated with vOTU genes, as depicted in Figure 3C. With respect to the biological processes, the vOTU genes predominantly participate in metabolism, cellular growth and development, and interspecific communication. From a molecular function standpoint, these genes are principally associated with catalytic activity, molecular binding, and small-molecule activity. As for cellular localization, the vOTU genes are predominantly localized within viruses and cellular environments. Synthesizing insights from both the KEGG and COG databases, it is evident that the vOTU genes situated within metabolic pathways often manifest catalytic functions through small-molecule interactions. To delve deeper into the metabolic function of vOTU genes and their influence on Tibetan pigs across various age groups, we conducted a statistical analysis on the coverage of each vOTU across the three cohorts using the Kruskal–Wallis test. For those samples exhibiting pronounced differences, pairwise significance was assessed using Dunn’s test. Based on this methodology and as presented in Appendix A, the vOTUs can be delineated into six distinct categories: a. vOTUs that manifest exclusively at a specific age stage of Tibetan pigs; b. vOTUs that are present across all three growth stages without substantial variations in coverage; c. vOTUs whose coverage escalates significantly as Tibetan pigs grow; d. vOTUs with a marked increase in coverage during the pigs’ growth, subsequently followed by a significant decrease; e. vOTUs that consistently demonstrate a notable decrease in coverage throughout the growth phases of Tibetan pigs; and f. vOTUs that initially show a pronounced decrease in coverage but subsequently exhibit a significant upsurge as the pigs mature. From this categorization, a total of 40,435 vOTUs were apportioned across the six categories. Notably, aside from the category with stable coverage, category ‘c’—representing continuous growth—has the most significant representation (Appendix A). Secondly, an additional annotation analysis was performed on the grouped vOTUs, and the calculated enrichment levels and q-values were retained, as depicted in Figure 3D,E.

In Figure 3D, regarding the selected results from the Gene Ontology (GO) level 3 terms, significant enrichment was observed in group a (manifesting exclusively at a specific age) for terms such as ‘killing of cells of other organism,’ ‘cell killing,’ ‘modulation of process of other organism’, and ‘immune response’. The host cell surface binding functions were enriched in the growth followed by decline group (group d), indicating a putative upregulation of viral invasion function after weaning. In the continuous decline group (group e), enrichment in the function ‘viral terminase’ complex was noted. The decline followed by rise group showed enrichment in the ‘cell aggregation’ and ‘cell adhesion’ functions.

In Figure 3E, the selected results from the Gene Ontology (GO) level 4 terms demonstrated notable enrichment (*p* < 0.05) regarding the terms related to ‘response to salt’ and ‘response to heat’ in the continuous growth group (group c), suggesting that viruses exhibiting sustained growth across the growth stages possess better environmental tolerance. ‘Aggregation of unicellular organisms’ and ‘cell-substrate adhesion’ were enriched in the decline followed by rise group (group f).

### 3.7. Auxiliary Metabolic Genes Identification from Tibetan Pig Gut Virome

DRAM-v software v1.5.0 was used based on annotations from the KEGG database, and five putative vAMGs were identified from the thirty-eight fragments (Figure 4A,B and Appendix A). After calculating the cumulative TPM to determine the relative abundance for each sample, the sum value was used to represent the vAMGs’ abundance across different growth periods. The proportions of the five vAMGs are as follows: *DNMT1* (31%), *queD* (23%), *GCH1* (23%), *queC* (16%), and *NAMPT* (5%) (Figure 4A). It can be visualized from the results that the lactation group has more abundant vAMGs than the nursery and fattening groups. Profiles of vAMGs, including *queD*, *GCH1*, and *queC*, were mainly retained in the lactation stage, while *DNMT1* presents at certain levels at each stage. Meanwhile, four putative metabolic pathways were found to be related to the vAMGs, including cysteine and methionine metabolism, folate biosynthesis, cofactor biosynthesis, and nicotinate and nicotinamide metabolism (Figure 4B). A similar decreasing trend with age growth was observed.

The potential origins of vAMGs can be traced through the analysis of vAMG phylogenetic trees (Figure 4C). The host of *DNMT1* associated with cysteine and methionine metabolism was predicted. Phylogenetic analysis showed that viral *DNMT1* from the Tibet sample clustered with *DNMT1* from *Phycisphaerales* and *Flavobacteri*, indicating that the origin of this vAMG may be traced back to phages infecting these bacteria.

### 3.8. Phage Lifestyle Identification Results

To ensure the accuracy of our predictions, two software programs, PHATYP and BACPHLIP, were employed. After data pre-processing, 27,945 vOTUs identified as phage sequences by geNomad were entered into the PHATYP software, of which 5079 vOTUs were further identified as complete by checkV and entered into the BACPHLIP software. As illustrated in Figure 5, despite the differences in the proportion of temperate phages between the two software outputs, similar trends were found between the samples. The lytic phages found dominated the composition in both outputs, and the summed TPM value of the lytic phages was significantly higher than that of the temperate phages within the three groups (*p* < 0.001). A comparison of the TPM totals for each group revealed that the mean and variance of the abundance of temperate phages in the nursery group were smaller than those in the lactation group, and the distributions of the conservation and fattening groups were roughly the same. The Kruskal–Wallis test yielded an insignificant *p*-value (*p* = 0.1212).

## 4. Discussion

Research on the genomic characterization of species in unique environments has been a longstanding focus. A recent study on Tibetan humans and Tibetan pigs compared samples from different altitudes using 16S-based diversity analysis, revealing unique gut bacteriomes and host adaptation events in high-altitude conditions [8]. A previous study conducted metatranscriptomic sequencing of fecal samples from 56 bird species and 91 small mammals, including shrews, rats, hamsters, and pikas, to characterize the RNA virome. In this study, 12 vertebrate RNA viruses were identified, exhibiting less than 40% amino acid identity to known viruses [50]. Another study focused on the viromes of herbivorous animals, including camels, cattle, and donkeys, in the northwest plateau of China, revealing diversity differences among regions and species [51]. These findings underscore the potential for Tibetan pigs in the Qinghai–Tibetan Plateau region to possess distinct intestinal virome compositions compared to domestic pigs in lowland areas. Intestinal virome analysis has been applied to Tibetan pig diarrhea samples [52]. Moreover, this study is the first to investigate the composition and function of gut virome communities in normal Tibetan pigs at different developmental stages. A total of 15 samples were collected from the lactation, nursery, and fattening stages, generating 150 GB of sequencing data. Our analysis identified the primary viral community, including *Microviridae*, *Petitvirales* unclassified family, and *Cirlivirals* unclassified family, while suggesting potential major hosts, such as *Lactobacillus*, *Streptococcus*, and *Flavobacterium*. Comparisons with other Chinese studies revealed variations in viral communities. For instance, a study in Guangxi Province on neonatal domestic piglets identified *Siphoviridae* (*Caudoviricetes*), followed by *Myoviridae* and *Podoviridae*, as the major viral community [53]. In a separate study in Hubei Province (Duroc x Landrace x Yorkshire), *Podoviridae* was found to be the most dominant viral community, followed by *Siphoviridae* (*Caudoviricetes*), and then *Myoviridae* [15]. In a study from Shanxi, the characterization results varied across different regions, with *Siphoviridae*, *Podoviridae*, and *Microviridae* being the primary communities, respectively, in the domestic pig gut [44]. Meanwhile, *Podoviridae* and *Caudoviricetes* ranked lower in Tibetan pigs compared to domestic pigs. 

Environmental factors, such as dietary patterns and living conditions, significantly influence the gut microbiome relative to the host’s genetic background [54]. Despite the debate over whether geography impacts the composition of viral communities in the pig gut [44,55], *Microviridae* were not the predominant community in lower-altitude areas [15,53]. Our findings in Tibetan pigs revealed that *Microviridae* were consistently the predominant viral community at all the stages of development in Linzhi, Xizang, PR China (altitude ≈ 3000 m) (Figure 1B). Interestingly, similar patterns have been reported in northern pig farms with high altitude (between 1000 and 2200 m) in Shanxi Province, PR China, where *Microviridae* constituted a significant proportion of the gut virome [44], suggesting that the abundance of the *Microviridae* family in the intestinal community may be associated with geographical locations with high altitude. 

Viruses are considered to be important components of the gut microbiome, and recent studies have identified the interactions between viruses and bacteria that affect the biological health of the host through functional annotation [56,57,58]. Through our functional annotation and enrichment of the six vOTUs change patterns, we found that, among the constantly increasing vOTUs groups, there are some enriched functions related to viruses’ adaptability to the environment, such as salt response and heat response. The genes enriched in this category, including hunger-related gene *groEL* and housekeeping gene *rpoD*, have been shown to have increased resistance to heat treatment and oxidative stress in *Campylobacter jejuni*. Meanwhile, we found that some functional terms contributing to viral invasion peaked at the nursery stage, such as ‘host cell surface binding’. This suggests the possibility of increased viral aggressiveness after weaning.

Viral-encoded vAMGs have demonstrated their influence on host metabolic processes and have been a topic of investigation in both animal gut and environmental systems [54,59]. This study represents the first attempt to characterize vAMGs across various growth stages of Tibetan pigs through fecal virome analysis. As for the lactation stage, the vAMGs involved *queD*, *queC*, and *GCH1*, whose functions, related to folate and cofactors biosynthesis, are significantly higher than those of the other two stages. The folate biosynthesis pathway associated with *queD*, *queC*, and *GCH1* contributes to the de novo synthesis pathway of queuosine (Que) in the process of tRNA biosynthesis [60]. Studies suggest that phages and their hosts may have complementary Que biosynthetic pathways, potentially enhancing translation efficiency through tRNA synthesis [61], suggesting that vAMGs involving *queD*, *queC*, and *GCH1* contribute to the successful colonization of their host by phage–bacteria mutualism during the lactation period. 

During the nursery and fattening stages, the contribution of many types of vAMGs decreased, leaving only *DNMT1* at a high level. *DNMT1* has been reported to participate in the protection of phages from the restriction–modification system [3,4], which are concurrently engaged in the synthesis of S-adenosyl-L-homocysteine (SAH) in bacteria. Interestingly, a comparable distribution was observed in human research where viral-encoded *DNMT1* and *MetK* were found to be highly prevalent in centenarians, contributing up to 50% of the combined viral and bacterial abundance [17], which may contribute as a factor in promoting health. 

Weaning is regarded as a challenge for intestinal microbial community homeostasis [62]. Various investigations have demonstrated that vAMGs play a crucial role in promoting the host’s adaptation to the environment [63,64], but studies on vAMG differences at various growth stages are scant. Notably, a substantial decrease in the abundance of most vAMGs from the lactation to the nursery period in this study (Figure 3C) suggests that the weaning process might have a significant impact on the composition of the microbiome in Tibetan pigs, and weaning represents a transition from liquid to solid food [65], so this nutritional shift could be a key factor causing changes in the microbial genomes.

Phages capable of undergoing lysogenic replication are referred to as temperate phages [66]. The distinction between the lysogenic cycle and the lytic cycle lies in the integration of the phage genome into the bacterial genome during its life cycle [67]. Metagenomic studies have identified temperate bacteriophages as prevalent and adaptable components within the viromes of both the human gut and deep ocean [17,19]. In this study, a coexistence analysis of lysis–lysogeny was conducted for the first time on the fecal virome obtained from Tibetan pigs. The results showed that viruses preferably adopt lytic lifestyles from the lactation stage to the fattening stage. 

The prevailing view is that phages tend to favor a lysogenic lifestyle in harsh host environments [68]. However, this study showed that viruses preferentially adopt a lytic lifestyle from the lactation to the fattening stage, suggesting that phages are more active in their invasive behavior. This finding supports the increased viral aggressiveness observed in functional annotations. In this study, Lactobacillus was regarded as the putative primary host, which is a typical probiotic involved in the digestion of protein and fiber feeds [69]. Research indicates that, during the succession of the intestinal microbiota in piglets, Lactobacillus gradually assumes a dominant position in the ileum as the piglets develop [70]. Other studies have demonstrated that host abundances can impact the invasibility of phages; higher densities of susceptible hosts elevate the likelihood of phage particles released through lysis encountering and infecting new hosts [71,72,73]. Combining the information above, we propose that the active state of bacteriophages may be attributed to the dominant development of the primary host in the intestinal tract.

In recent years, diarrheal diseases caused by diarrheagenic *Escherichia coli* (DEC) have become a significant concern in Tibetan pig farming. Z Cao et al. collected feces of Tibetan piglets from the Nyingchi area and demonstrated a 41.3% isolation rate of DEC from Tibetan pigs [74]. A previous study investigated the adverse effects of DEC on the gut microbiota of Tibetan piglets, but research on viromes in the same condition remains limited [75]. Azadeh Vahedi et al. demonstrated the therapeutic potential of phage therapy through in vitro and in vivo experiments on DEC [76]. This study presents a virome analysis based on healthy samples, providing baseline data for future virome studies in Tibetan pigs, especially under specific pathological conditions. It should be noted that this study does not provide a complete analysis of the virome as RNA viruses were not included, and no typical enteropathogenic viruses were detected. Additionally, it is important to recognize that the abundance of phage nucleic acid in fecal samples is influenced by the abundance of bacteria. A high concentration of bacterial nucleic acid, even following viral enrichment, can hinder the detection of viral nucleic acid.

## 5. Conclusions

This study provides an in-depth analysis of the gut virome in Tibetan pigs at different developmental stages, highlighting the unique adaptation to the high-altitude Tibetan Plateau. The consistent presence of *Microviridae* phages across all stages suggests a potential link with the geographical and environmental conditions. In the vOTUs groups exhibiting continual growth, functional annotation emphasizes the viruses’ adaptability to environmental challenges like heat and salt. This suggests an evolutionary adaptation in response to the demanding high-altitude conditions. The discovery of distinct patterns in viral-encoded vAMGs across different growth stages, especially in relation to crucial sulfur metabolic pathways, underscores the dynamic interplay between the gut virome and the host’s developmental and environmental context. Interestingly, the study also observes a predominant lytic lifestyle in these viruses, offering a new perspective on phage–host interactions in harsh environments. These findings contribute to our understanding of the microbial ecology of Tibetan pigs and could inform further studies in similar extreme environments.

## Figures and Tables

**Figure 1 viruses-16-00606-f001:**
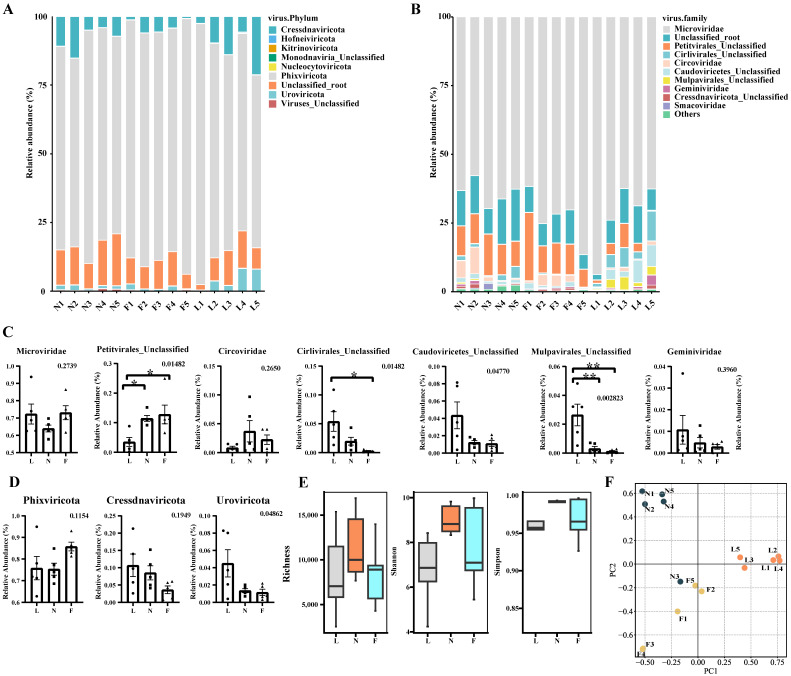
Results of DNA virus taxonomy composition across three developmental stages. (**A**,**B**) Relative abundance distribution at the virus phylum level and virus family level. (**C**) One-way analysis of variance (ANOVA) for the seven most abundant virus families at the family level. The shapes of the points vary according to three age groups. The number in the top right corner represents the *p*-value from one-way ANOVA. When the difference is significant (*p* < 0.05), pairwise *t*-tests are performed between groups; (*): *p*_t_ < 0.05, (**): *p*_t_ < 0.005 (**D**) One-way analysis of variance (ANOVA) for the seven most abundant virus phylumat the phylum level. (**E**) Alpha diversity analysis. Different colors represent data from different age groups. (**F**) Principal Coordinates Analysis (PCoA) at the phylum level. Different colors represent data from different age groups.

**Figure 2 viruses-16-00606-f002:**
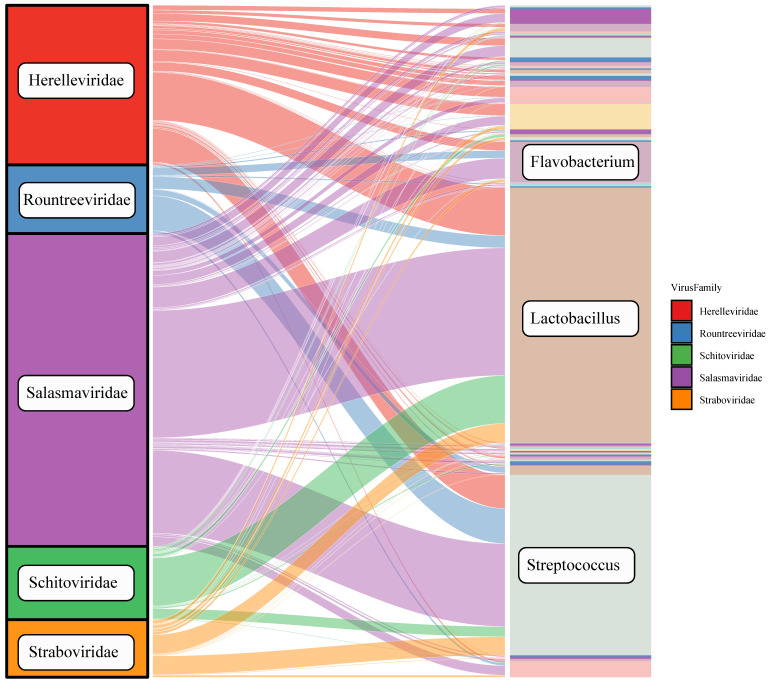
Sanky plot of the host prediction results from PhaBOX. On the left are viral families, and on the right are predicted host genera at the genus level.

**Figure 3 viruses-16-00606-f003:**
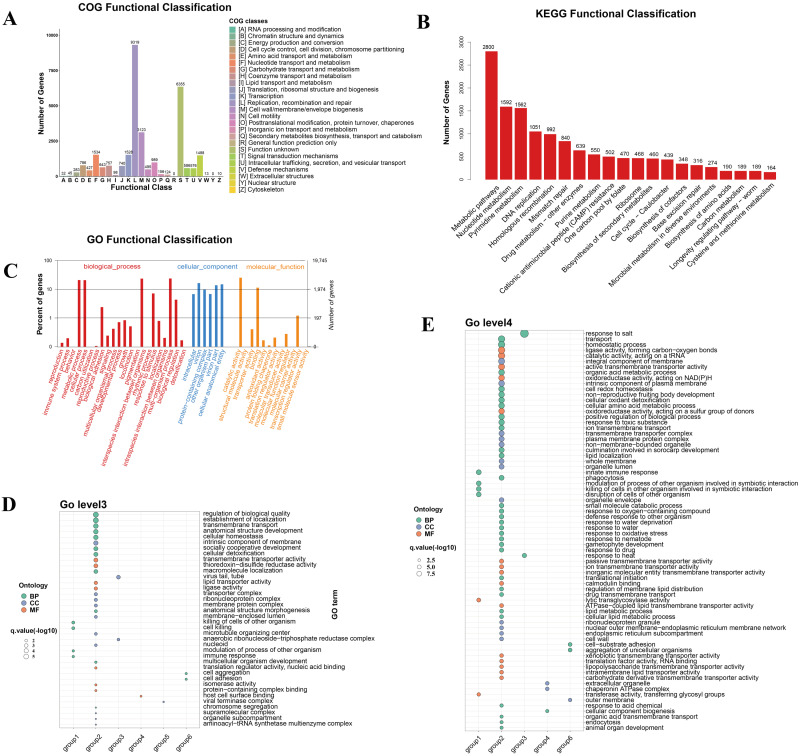
Functional annotation analysis of intestinal virome. (**A**) Result of Cluster of Orthologous Groups (COG) functional annotation. (**B**) Result of Kyoto Encyclopedia of Genes and Genomes (KEGG) functional annotation. (**C**) Annotation result at the level 2 of Gene Ontology (GO). (**D**,**E**) Result of GO enrichment based on the patterns of vOTUs categorized into six subgroups. Background genes include all genes, and foreground genes consist of genes within each subgroup. In Level 4, group e did not show a significant presence.

**Figure 4 viruses-16-00606-f004:**
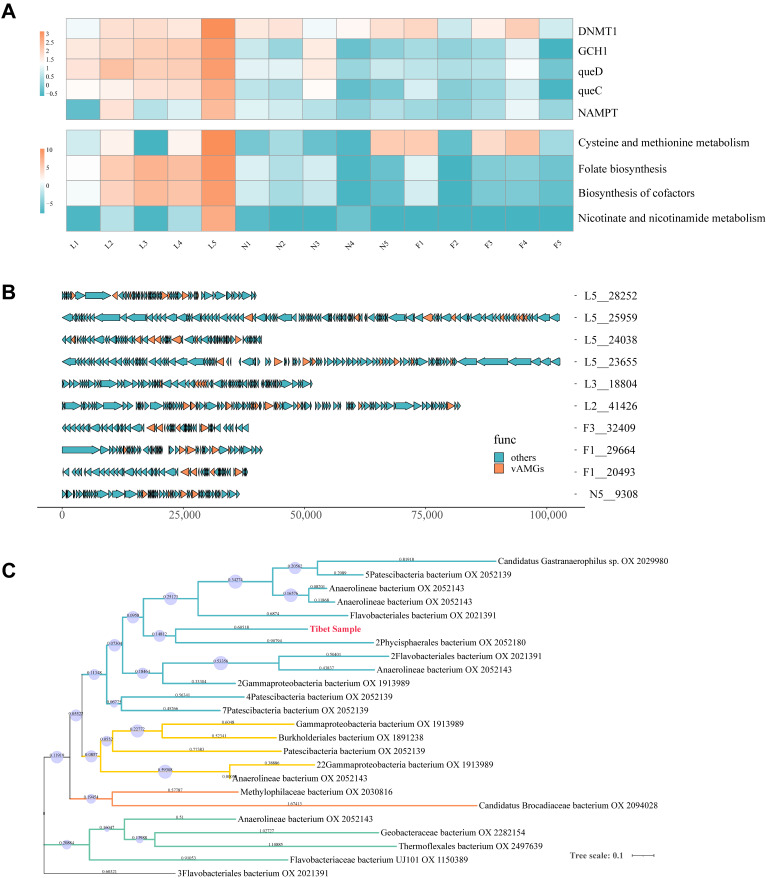
Comparative analysis of vAMGs with phylogeny. (**A**) Heatmap displaying the standardized total Transcript Per Million (TPM) sum for each sample entry. Heatmap color intensity after summing and applying log10 to all values. Upper panel represents metabolic annotated genes by KEGG, while lower panel represents annotated pathways. (**B**) Positions of vAMGs in the virus gene, exemplified by 10 vOTUs. (**C**) Phylogenetic analysis of the *DNMT1* obtained in this study compared to others from bacteria. The results are categorized into clades represented by four different colors, with the Tibet sample labeled in red. The image was modified by iTOL.

**Figure 5 viruses-16-00606-f005:**
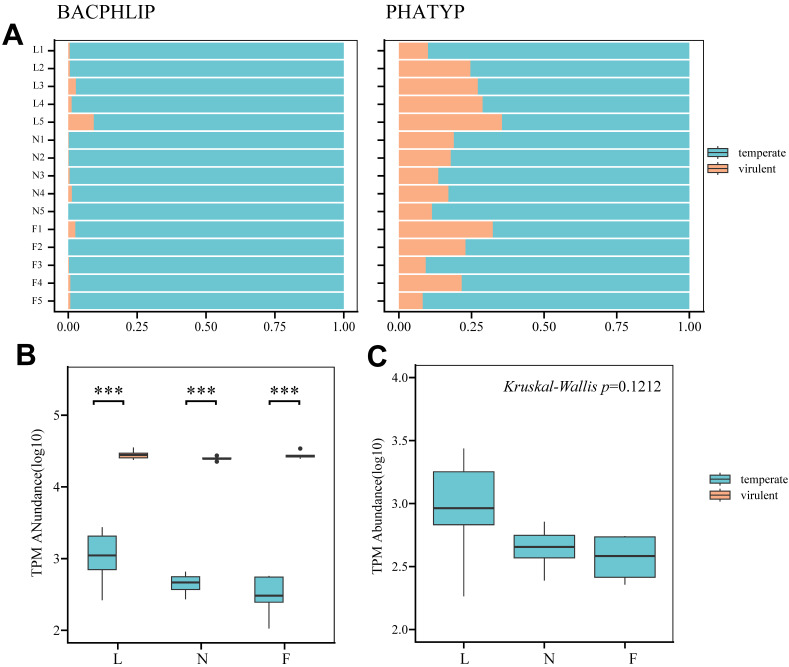
Results of lysogenic and lytic lifestyles prediction and abundance calculation. (**A**) Predicted phage lifestyles by BACPHLIP and PHATYP. The x-axis represents the proportion of vOTUs count relative to the total. (**B**) Box plots for each development stage representing the sum of log10-transformed TPM values of vOTUs in 5 samples. *t*-tests are employed to detect differences between each group of temperate and virulent. ***: *p* < 0.001 (**C**) Box plots of lytic abundance across three developmental stages. The box plots display the distribution of lytic abundance across three developmental stages. The Kruskal–Wallis test result, with a *p*-value of 0.1212, indicates no significant difference in lytic abundance among the stages (*p* > 0.05). L: lactation group; N: nursery group; F: fattening group.

## Data Availability

The datasets supporting the conclusions of this article have been submitted to the NCBI Sequence Read Archive (SRA) repository under BioProject PRJNA1073687.

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
