# Peer review of "Developmental Dynamics of the Gut Virome in Tibetan Pigs at High Altitude: A Metagenomic Perspective across Age Groups"

_viruses, 2024, doi:10.3390/v16040606_

Round 1

Reviewer 1 Report

Comments and Suggestions for Authors

The manuscript presents a deep analysis of phage genetics at different stages of life of Tibetan pigs. 

Some comments for clarification and improvement of the manuscript:

Lines 67-68: This sentence is not clear, please elaborate a little more. It reads like all intestinal viruses are primarily porcine rotaviruses. 

Line 126: For how long the samples were stored at -20 before processing? It can affect the viability of viral RNA. 

Line 145: What is the NGS library prep method? it is an essential piece of this manuscript and should be described. 

It should be clearly mentioned in the manuscript that this is not an analysis of the complete virome, since viruses that infect vertebrates were not described or not detected in this study. Also, it should be acknowledged that the abundance of phage nucleic acid is due to the abundance of bacteria in fecal samples. The high concentration of bacterial nucleic acid, even after viral enrichment, can impair the detection of viral nucleic acid.

Author Response

For research article

Response to Reviewer 1 Comments

1. Summary

2. Questions for General Evaluation

Reviewer’s Evaluation

Response and Revisions

Does the introduction provide sufficient background and include all relevant references?

Yes

Are all the cited references relevant to the research?

Yes

Is the research design appropriate?

Yes

Are the methods adequately described?

Can be improved

We agree with this comment

Are the results clearly presented?

Yes

Are the conclusions supported by the results?

Can be improved

We agree with this comment

3. Point-by-point response to Comments and Suggestions for Authors

Comments 1:

Lines 67-68: This sentence is not clear, please elaborate a little more. It reads like all intestinal viruses are primarily porcine rotaviruses.

Response 1: Thank you for your review and valuable suggestions regarding our manuscript! We sincerely apologize for the lack of clarity in the sentence you pointed out. The intended meaning was to describe the results of a study on a condition of pathogen infections. However, we realize that the background of the research was not adequately explained in the manuscript. We have now addressed this issue by revising the sentence, which can be found in line 70-72 of the manuscript. The revised content is as follows:

line70-72

“In a study focusing the virome of wasting piglets aged 0 to 9 weeks, it was discovered that weaning facilitated the transition of pathogenic viruses from enterovirus G and rotaviruses to porcine sapovirus.”

Comments 2:

Line 126: For how long the samples were stored at -20 before processing? It can affect the viability of viral RNA.

Response 2: Thank you for your insightful comment. We are sorry for the oversight in not providing sufficient detail storage information. Throughout the experiment, the samples were stored at -20°C for a period of one week. We have now included a more comprehensive description of the sample storage process in the manuscript. This clarification can be found in line 130 of the article.

line 130

“Fecal samples were initially stored at -20°C for one week.”

Comments 3:

Line 145: What is the NGS library prep method? it is an essential piece of this manuscript and should be described.

Response 3: Thank you for your comment. We apologize for the oversight regarding the description of the NGS library preparation method. We recognize that this information is crucial for understanding our study. We have now added a detailed description of the NGS library preparation method to the manuscript. This addition can be found in line 151-157 of the revised manuscript.

line 151-157

“Metagenomic libraries were prepared following TruSeqTM Nano DNA sample preparation Kit from Illumina (San Diego, CA), using 1ug of total DNA. DNA end re-pair, A-base addition and ligation of the Illumina-indexed adaptors were performed according to Illumina’s protocol. Libraries were then size selected for DNA target fragments of ~400bp on 2% Low Range Ultra Agarose followed by PCR amplified using Phusion DNA polymerase (NEB) for 15 PCR cycles. All samples were sequenced by the Illumina NovaSeq 6000 platform (150bp*2, Shanghai Biozeron Biotechnology Co., Ltd, Shanghai, China).” was added.

Comments 4:

It should be clearly mentioned in the manuscript that this is not an analysis of the complete virome, since viruses that infect vertebrates were not described or not detected in this study. Also, it should be acknowledged that the abundance of phage nucleic acid is due to the abundance of bacteria in fecal samples. The high concentration of bacterial nucleic acid, even after viral enrichment, can impair the detection of viral nucleic acid.

Response 4:

Thank you for your constructive comment. We appreciate your suggestion, as it will enhance the logical coherence of our study. This clarification has been added in the last part of the discussion section.

line 556-561

“It should be noted that this study does not provide a complete analysis of the virome, as RNA viruses were not included, and no typical enteropathogenic viruses were detected. Additionally, it is important to recognize that the abundance of phage nucleic acid in fecal samples is influenced by the abundance of bacteria. The high concentration of bacterial nucleic acid, even following viral enrichment, can hinder the detection of viral nucleic acid.”

Reviewer 2 Report

Comments and Suggestions for Authors

The manuscript entitled “Developmental Dynamics of the Gut Virome in Tibetan Pigs at  High-Altitude: A Metagenomic Perspective Across Age Groups” is well written and well-structured in each of its sections by the Authors.

The study of viroma is interesting both in relation to altitude and generally extreme conditions and in consideration of its co-development with the intestinal flora and could open up interesting prospects if applied to other production animals as well.

I would suggest that the authors include some consideration of this either in the introduction or in the discussion stage perhaps with reference to lagomorphs, other rodents, camelids or even small and graded ruminants intended for milk production while also advancing some hypotheses.

In contrast, the experimental plan devised and conducted by the researchers is absolutely faultless as is the detailed statistical analysis of the results obtained.

I would suggest that the authors also include a correlation analysis between viroma change and weight gain, provided they have this data available.

Moreover, in discussion section, it would be interesting to make more specific considerations regarding diarrheagenic escherichia coli (DEC), which is an important pathology of Tibetan pigs.

Author Response

For research article

Response to Reviewer 2 Comments

1. Summary

2. Questions for General Evaluation

Reviewer’s Evaluation

Response and Revisions

Does the introduction provide sufficient background and include all relevant references?

Can be improved

We agree with this comment

Are all the cited references relevant to the research?

Yes

Is the research design appropriate?

Yes

Are the methods adequately described?

Yes

Are the results clearly presented?

Yes

Are the conclusions supported by the results?

Can be improved

We agree with this comment

3. Point-by-point response to Comments and Suggestions for Authors

Comments 1:

The manuscript entitled “Developmental Dynamics of the Gut Virome in Tibetan Pigs at  High-Altitude: A Metagenomic Perspective Across Age Groups” is well written and well-structured in each of its sections by the Authors.

The study of viroma is interesting both in relation to altitude and generally extreme conditions and in consideration of its co-development with the intestinal flora and could open up interesting prospects if applied to other production animals as well.

I would suggest that the authors include some consideration of this either in the introduction or in the discussion stage perhaps with reference to lagomorphs, other rodents, camelids or even small and graded ruminants intended for milk production while also advancing some hypotheses.

Response 1: Thank you for your review and valuable suggestions regarding our manuscript! We greatly appreciate your input. We fully agree with your suggestion and have made revisions to the manuscript accordingly. In response to your comment, we have included a discussion of the association between the gut virome and different animal species, with specific references to lagomorphs, other rodents, camelids, and small ruminants intended for milk production. These additional discussions have been incorporated into the relevant section of the manuscript, specifically in paragraph line 451-459 of the discussion.

line 451-459

“A previous study conducted metatranscriptomic sequencing of fecal samples from 56 bird species and 91 small mammals, including shrews, rats, hamsters, and pikas, to characterize the RNA virome. In this study, 12 vertebrate RNA viruses were identified, exhibiting less than 40% amino acid identity to known viruses. Another study focused on the viromes of herbivorous animals, including camels, cattle, and donkeys, in the northwest plateau of China, revealing diversity differences among regions and species. These findings underscore the potential for Tibetan pigs in the Qinghai-Tibetan Plateau region to possess distinct intestinal virome compositions compared to domestic pigs in lowland areas.”

Comments 2:

In contrast, the experimental plan devised and conducted by the researchers is absolutely faultless as is the detailed statistical analysis of the results obtained.

I would suggest that the authors also include a correlation analysis between viroma change and weight gain, provided they have this data available.

Response 2: Thank you for your kind words regarding the experimental design and statistical analysis of our study. We appreciate your positive feedback. Regarding your suggestion to include a correlation analysis between virome changes and weight gain, we feel sorry that the detailed weight data were not collected during the course of the experiment. However, we acknowledge the importance of this correlation and will certainly consider incorporating it into our future research endeavors. Your insight will undoubtedly guide us in enhancing the comprehensiveness of our investigations.

Comments 3:

Moreover, in discussion section, it would be interesting to make more specific considerations regarding diarrheagenic escherichia coli (DEC), which is an important pathology of Tibetan pigs.

Response 3: Thank you for your insightful suggestion regarding the discussion section of our manuscript. We agree that discussing diarrheagenic Escherichia coli (DEC) would be highly relevant, as it is an important pathology in Tibetan pig farming. After reviewing relevant literature, we have incorporated a discussion on DEC infection into the final paragraph of the discussion section. We believe this addition provides valuable insights into the broader context of our study. We sincerely appreciate your recommendation, as it has helped enrich the depth of our discussion.

line 547-553

“In recent years, diarrheal diseases caused by diarrheagenic escherichia coli (DEC) have become a significant concern in Tibetan pig farming. Z Cao et al. collected feces of Tibetan piglets from Nyingchi area and demonstrated a 41.3% isolation rate of DEC from Tibetan pigs. Previous study had investigated the adverse effects of DEC on gut microbiota of Tibetan piglets, but research on virome in the same condition remains limited. Azadeh Vahedi et al. demonstrated the therapeutic potential of phage therapy through in vitro and in vivo experiments on DEC. This study presents a virome analysis based on healthy samples, providing baseline data for future virome studies in Tibetan pigs, especially under specific pathological conditions.”
